# Role of Oxidative Stress in the Occurrence, Development, and Treatment of Breast Cancer

**DOI:** 10.3390/antiox14010104

**Published:** 2025-01-17

**Authors:** Rui Dong, Jing Wang, Ruiqi Guan, Jianwei Sun, Ping Jin, Junling Shen

**Affiliations:** 1Yunnan Key Laboratory of Cell Metabolism and Diseases, Center for Life Sciences, School of Life Sciences, Yunnan University, Kunming 650091, China; ruidong970942@163.com (R.D.); wangj2025@126.com (J.W.); guanruiqiYXNU@163.com (R.G.); jwsun@ynu.edu.cn (J.S.); 2State Key Laboratory for Conservation and Utilization of Bio-Resources in Yunnan, Yunnan University, Kunming 650091, China; 3Key Laboratory of Tumor Immunological Prevention and Treatment of Yunnan Province, Kunming 650051, China

**Keywords:** oxidative stress, breast cancer, drug resistance, therapeutic strategies, reactive oxygen species (ROS)

## Abstract

Breast cancer is one of the most prevalent cancers worldwide. Recent studies have increasingly emphasized the role of oxidative stress in the initiation and progression of breast cancer. This article reviews how oxidative stress imbalance influences the occurrence and advancement of breast cancer, elucidating the intricate mechanisms through which reactive oxygen species (ROS) operate in this context and their potential therapeutic applications. By highlighting these critical insights, this review aims to enhance our understanding of oxidative stress as a potential target for innovative therapeutic strategies in the management of breast cancer.

## 1. Introduction

### 1.1. Research Status of Breast Cancer

Breast cancer is increasingly recognized as a collection of molecularly defined syndromes that arise within the same organ, characterized by various molecular subpopulations that exhibit distinct clinical behaviors, metastatic patterns, and sensitivities to existing therapies. Over the past fifty years, breast cancer research has evolved dramatically, leading to significant improvements in patient survival rates and a reduction in mortality. For instance, advancements in surgical techniques have allowed many women to be effectively treated with simple tumor resection and minimal lymph node surgery [1]. Currently, research is primarily focused on the development of new drugs, with the National Cancer Institute conducting extensive screening programs that evaluate tens of thousands of compounds across various in vitro models, including those with unclear molecular targets [2]. In 2023, numerous new treatments were introduced, featuring more effective options designed to achieve better outcomes with fewer interventions. These advancements in technology and methodology enhance treatment efficacy and provide patients with improved choices [3,4].

The regulation of oxidative stress is a crucial factor in tumor development and the response to anti-cancer treatment. The redox status of cancer cells often differs from that of normal cells, with many signaling pathways related to tumorigenesis involved in regulating reactive oxygen species (ROS) through direct or indirect metabolic mechanisms that influence tumor progression [5,6]. In the context of breast cancer occurrence and treatment responses, it is common for patients to use antioxidant supplements during breast cancer chemotherapy, radiation therapy, and/or hormone therapy [7]. However, the precise effects of oxidative stress on cancer initiation, progression, and treatment response require further investigation. A deeper understanding of the mechanisms underlying oxidative stress and their influence on the tumor microenvironment (TME) may provide novel insights and strategies for breast cancer treatment. As research advances, more effective treatment strategies are anticipated to emerge, ultimately improving patient prognosis. We used the ‘PubMed database’ to search, found literature related to ROS regulation of breast cancer, classified and summarized them, and wrote this review article.

### 1.2. Reactive Oxygen Species in Cells

The continuous reduction in molecular oxygen (O_2_) leads to the generation of various oxygen-containing species, including hydrogen peroxide (H_2_O_2_), superoxide anion (O_2_^−^), and hydroxyl radicals (^•^OH), collectively known as ROS. Among these species, hydroxyl radicals (^•^OH) are the most highly reactive, less stable, and most damaging to macromolecules compared to H_2_O_2_. H_2_O_2_ is regarded as the primary ROS involved in the redox regulation of biological activity [8,9]. ROS have two opposing properties. For example, signaling molecules that are beneficial in some cases and harmful oxidants in others [10]. While ROS play essential roles in cell metabolism, immune responses, and signaling pathways, excessive levels can result in cellular damage [11]. ROS have numerous important physiological effects on cells, including participation in cell signal transduction-activating transcription factors and regulating signal cascade-promoting cell proliferation, inducing cell differentiation, affecting cell death and survival, and regulating the antioxidant defense system [12]. Under normal physiological conditions, cells produce a variety of antioxidants, such as glutathione (GSH), vitamin C, and vitamin E, which help neutralize ROS and maintain redox balance. However, when antioxidant defenses are insufficient to counterbalance ROS production, oxidative stress can compromise cellular function [13,14]. ROS can be generated from various cellular pathways, including NADPH oxidase (NOX) [15,16], the mitochondrial respiratory chain [17,18,19], endoplasmic reticulum (ER) stress [20], peroxisomes [21], and external stimuli [22,23].

NOX proteins are a class of membrane-associated enzymes responsible for transferring electrons, with oxygen typically serving as the electron acceptor, leading to superoxide production [24]. NOX was first identified in neutrophils and macrophages, where these cells generate a significant amount of ROS during the inflammatory response, forming the body’s primary defense against pathogens. The human homologs of the NOX family include NOX1, NOX3, NOX4, NOX5, DUOX1, and DUOX2 [25], each mediating ROS involved in various physiological functions, such as immune host defense and multicellular signaling pathways.

Mitochondria are critical sites for cell energy production. During the oxidative phosphorylation process, complexes I and III of the mitochondrial respiratory chain generate O_2_^−^, which is subsequently converted into H_2_O_2_ by superoxide dismutase (SOD) [26,27] and various peroxidases, including glutathione peroxidase (GSH-Px) and peroxiredoxins [28].

ER serves both as a production site for H_2_O_2_ and as a hub for H_2_O_2_^−^ mediated signaling events. The lumen of the ER is a significant source of H_2_O_2_ generated during disulfide bond formation in protein folding. These mechanisms lead to fluxes of O_2_^−^ and H_2_O_2_, influencing processes like protein translation and proapoptotic signaling pathways. The diffusivity of H_2_O_2_ allows for ROS production both within cells and in the extracellular space, impacting molecules located in the ER [20,29,30]. Sustained ER stress and protein misfolding can trigger ROS cascades, playing important roles in the pathogenesis of various human diseases.

Peroxisomes are eukaryotic organelles involved in fatty acid α-oxidation, very long-chain fatty acid (VLCFA) β-oxidation, purine metabolism, and the biosynthesis of glycerolipids and bile acids. Their major enzyme components include catalase and flavin oxidase, which generate H_2_O_2_ [31]. Peroxisomes are central organelles in the dynamic cycle of ROS generation and clearance [32]. To mitigate the damaging effects of ROS, peroxisomes contain various antioxidant enzymes, including catalase, which reduces H_2_O_2_ to water. A disruption in the balance between ROS production and clearance within the peroxisome can lead to oxidative stress, which is associated with several age-related human diseases, including diabetes, cancer, and neurological diseases [33,34].

Both cellular metabolism and environmental factors significantly influence ROS generation. Normal metabolic processes, such as oxidative reactions during fatty acid metabolism and amino acid catabolism, produce ROS as byproducts, contributing to the overall cellular ROS pool [35]. Additionally, external factors like ultraviolet radiation, pollutants, and tobacco smoke can increase ROS levels, inducing oxidative stress and exacerbating cellular damage [23]. Understanding the mechanisms of ROS generation and their roles in various cellular contexts is essential for elucidating their contributions to numerous disease states, including breast cancer.

## 2. Relationship Between Oxidative Stress and Breast Cancer

Oxidative stress is known to play various functions in non-cancer diseases [36,37], and it is also significant in different stages of breast cancer development. As tumors progress, fluctuations in oxidative stress levels can influence tumor growth, metastasis, and responses to treatment [38]. Research indicates that low antioxidant levels and increased oxidative stress may be detectable even before clinical symptoms of tumors emerge [39,40]. In the early stages of tumorigenesis, oxidative stress can cause DNA damage and induce mutations. ROS can modify genetic material by oxidizing DNA bases, contributing to the malignant transformation of cells [41]. In response to oxidative stress, cancer cells can activate signaling pathways that promote their survival and proliferation, including the PI3K/Akt and MAPK pathways, which are associated with cell growth and survival. The activation of these pathways allows tumor cells to endure the damaging effects of oxidative stress, facilitating their proliferation [39]. Moreover, oxidative stress can influence tumor development by affecting the TME. It often leads to increased inflammation, a significant factor in cancer progression. In this context, ROS produced by immune cells can enhance the proliferation and metastasis of tumor cells [42]. Additionally, studies have shown that during the metastatic process, tumor cells can adapt to oxidative stress by increasing NADPH levels, which may provide insights for developing new therapeutic approaches [43].

The onset, spread, and management of breast cancer are closely linked to the nuclear receptor family of transcription factors, including the estrogen receptor (ER), androgen receptor (AR), and progesterone receptor (PR) [44]. All of the main subtypes of breast cancer demonstrate interactions between nuclear receptor pathways and growth factor signaling, which are utilized both diagnostically and prognostically. There is increasing evidence that nuclear receptors and oxidative stress are interconnected in breast cancer. For instance, the activation of ER can elevate ROS production, contributing to oxidative stress [45]. Conversely, certain nuclear receptors, such as peroxisome proliferator-activated receptors (PPARs), may exert antioxidant effects by regulating genes involved in the oxidative stress response and cellular protection mechanisms [46]. This interaction creates a feedback loop in which nuclear receptor signaling can influence oxidative stress levels and vice versa, potentially affecting cancer cell survival and the response to treatment.

Overall, oxidative stress plays a complex and crucial role in tumorigenesis, influencing not only the initial stages of tumor formation but also the survival, proliferation, and microenvironment of tumor cells. A comprehensive understanding of the mechanisms underlying oxidative stress is essential for the prevention and treatment of breast cancer. Future research should focus on exploring effective strategies to regulate oxidative stress, potentially leading to novel therapeutic options for cancer treatment. The following sections summarize current findings on the role of oxidative stress in the onset and metastasis of breast cancer.

### 2.1. Excessive ROS Lead to DNA Damage

#### 2.1.1. ROS Can Cause DNA Replication Stress

Various exogenous and endogenous stimuli, including chemicals, radiation, free radicals, and topological changes, can damage DNA. The resulting alterations in replication fork processes, reduced replication fidelity, and DNA fragmentation due to stress are collectively referred to as replication stress (RS) [47,48] (Figure 1A). Abnormal levels of ROS can induce RS, which involves disruptions in DNA synthesis and abnormal replication fork processing [48,49]. DNA damage and RS are often more pronounced in precancerous and tumor cells compared to normal tissue, suggesting that RS contributes to genomic instability. When DNA replication is disrupted, allelic imbalance occurs at sites prone to DNA double-strand breaks (DSBs); thus, cancer development is associated with RS from its earliest stages, leading to DSBs, genomic instability, and selective pressure for *p53* mutations [50,51]. Oncogene activation further increases ROS levels, which can exacerbate RS [52,53].

ROS regulate the oxidative modification of dNTPs, impacting the gene mutation rate and ultimately contributing to cancer development [54]. They affect DNA synthesis efficiency by oxidizing dNTPs, which leads to RS [55,56]. Oxidatively modified DNA is prevalent in many human tissues, especially guanine (G), which is easily oxidized to 8-oxo guanine (8-OG) owing to its low redox potential [57,58].8-OG preferentially pairs with adenine instead of cytosine, resulting in GC→TA inversion mutation after replication [59,60]. Some studies indicate that the 8-oxo dGTP base can disrupt charge regulation during insertion, blocking DNA repair intermediates and causing further DNA damage [61]. Moreover, when 8-OG is present in the template strand of a protein-coding region, RNA polymerase II can become stalled, leading to truncated transcripts [62,63,64]. The oxidation of guanine also prevents the formation of Hoogsteen hydrogen bonds, hindering G-quadruplex formation and affecting DNA replication and transcription efficiency [65]. Peroxynitrite (ONO_2_^−^) is another oxidant that can damage DNA. Exposure to ONO_2_^−^ can result in broken DNA strands and base oxidation products, leading to deamination and oxidation of G and A [66]. Research has shown that the occurrence of G→T transitions in the p53 gene is linked to nitric oxide synthase expression levels [67]. Other oxidatively modified bases include formamide pyrimidine adducts of adenine and guanine, which cause GC→CG inversion [68]. Additionally, various modified bases such as 5-hydroxymethyl-thymidine, 5-hydroxymethyl-uracil, uracil glycol, cytosine glycol, and thymine glycol contribute to DNA damage [66], with thymidine pairing with adenine leading to C→T transition [69].

Furthermore, replication forks encountering DNA single-strand breaks (SSBs) generated by ROS and other factors can lead to replication fork run-off and the formation of one-ended DSBs [70] (Figure 1B). H_2_O_2_ is believed to interfere with replication fork progression by inducing replication fork turnover mediated by RAD51 recombinase and the DNA translocases ZRANB3, HLTF, and SMARCAL1 [71,72,73]. ROS can also disrupt replication fork progression by dissociating peroxiredoxin 2 oligomers (PRDX2), which forms a replicator-associated ROS sensor that binds to the fork accelerator TIMELESS. Elevated ROS levels cause the dissociation of PRDX2 and TIMELESS, slowing replication forks [74]. Additionally, the oxidation of bases by ROS presents a physical barrier to replication forks, leading to their disintegration at vulnerable genomic sites, resulting in DSBs and ultimately causing under- or over-replication of DNA [75]. While excessive oxidative stress is detrimental to normal cells, Liao H’s team have developed a replication stress nanoamplifier (RSNA) composed of catalytic FePt nanoparticles loaded with chemotherapeutic doxorubicin (DOX). This RSNA selectively exacerbates RS in cancer cells, promoting replication fork catastrophe. In cancer cells, RSNA converts excess ROS into oxygen, enhancing DNA damage caused by radiotherapy and increasing template damage, which hinders the coordinated progression of the replication fork and DOX, thereby overcoming the breast cancers’ resistance to chemotherapy [76]. Additionally, in cancer cells, the presence of DNA in the cytoplasm, resulting from replication fork collapse or ROS, can constitutively activate the cGAS-STING pathway, inducing chronic IFN-β expression. This enhances the recruitment of effector immune cells, and the acute response contributes to cell death [77].

#### 2.1.2. ROS Affect the DNA Damage Repair Response

When DNA damage occurs, cells activate a series of responses known as DNA damage responses (DDRs), which include the recognition of DNA damage, checkpoint activation, cell cycle arrest, repair mechanisms, apoptosis, and immune clearance, all of which work to maintain genome stability and integrity [78,79]. Early precursor disease tissue often shows markers of an activated DDR, which can delay or prevent the onset of cancer [79,80]. Additionally, ROS can significantly impact the DDR process (Figure 1C). In vertebrate cells, the DDR is primarily regulated by three closely related kinases: ataxia-telangiectasia mutated (ATM), ataxia telangiectasia and rad3–related (ATR), and DNA-dependent protein kinase (DNA-PK) [81]. ATM is rapidly recruited to sites of DSBs by the Mre11-Rad50-Nbs1 (MRN) complex, which acts as a DNA damage sensor. Once recruited, ATM phosphorylates histone H2A variant (H2AX) and mediator of DNA damage checkpoint protein 1(MDC1) initiate DNA damage repair [82,83].

A growing number of studies indicate a strong correlation between ATM and ROS oxidation. The oxidized form of ATM is a disulfide cross-linked dimer, which differs from the mechanism typically involved in DNA damage activation highlighting ATM’s role as a significant sensor of ROS in human cells [84,85]. The cofactor protein ATMIN (ATM interactor) is crucial for ATM’s redox response, mediating ATM activation in the presence of oxidative stress, thereby helping to reduce the DNA damage accumulation in the brain [86]. Furthermore, ATM is known to activate the pentose phosphate pathway to boost antioxidant production and facilitate DNA repair [87]. Conversely, ROS can also inactivate DNA repair enzymes through oxidation, contributing to increased genomic instability. For example, ROS generated by ionizing radiation can inactivate essential DNA repair proteins, including the MRN complex, DNA-PK, and DNA repair helicase such as xeroderma pigmentosum group D(XPD) and fanconi anemia complementation group J (FANCJ) [85,88,89]. Additionally, elevated ROS levels lead to degradation of the cell cycle phosphatase CDC25C [90]. The apurinic/apyrimidinic endonuclease (APE) serves as the central signal transduction point that initiates certain repair and transcription pathways. Increased ROS activate apurinic/apyrimidinic endonuclease 1 (APE1), enhancing the repair of cytotoxic DNA damage [91]. ADP–ribose polymerase 1/2 (PARP1/2) plays critical and overlapping roles in major DNA repair pathways and in maintaining genome stability. PARP inhibitors have been approved for the treatment of breast cancer susceptibility gene 1/2 (*BRCA1/2*)-mutated ovarian cancer and breast cancer [92]. Research by Ren Y has shown that the natural compound alantolactone (ATL) can inhibit thioredoxin reductase, leading to an accumulation of ROS in cancer cells, this accumulation oxidizes damaged DNA, activates PARP in cancer cells, and causes further oxidative damage to DNA [93].

#### 2.1.3. ROS and DNA Damage in Breast Cancer Tumor Cells

Compared to normal cells, tumor cells often harbor genetic alterations that result in sustained and elevated production of ROS [94]. These ROS can cause oxidative DNA damage and contribute to tumor growth in several ways including inducing DNA damage, promoting genomic instability and ultimately reprogramming cancer cell metabolism [95]. Oncogene-induced ROS are considered crucial mediators of proliferative signals. Research by M Ogrunc et al. reported that ROS function as signaling molecules that promote cell proliferation and drive abnormal cell proliferation triggered by oncogenes. This excessive cell proliferation can lead to the activation of the DDR [96]. For instance, the overexpression of *c-MYC* induces ROS production and causes DNA damage [97]. Comparative studies of *RAS* and *MYC* have indicated that both oncogenes cause shifts in cellular metabolic patterns, resulting in varying degrees of oxidative stress and RS [52]. Additionally, Thomas J Hayman reported that the protein stimulator of interferon genes (STING) regulates ROS production at the transcriptional level; the loss of STING disrupts ROS homeostasis, reducing DNA damage and contributing to treatment resistance [98]. HuR, an mRNA-binding protein, is frequently overexpressed in cancer cells and is associated with poor prognosis and treatment resistance. Molecular studies suggest that increased ROS production and the inhibition of thioredoxin reductase (TrxR) in HuR-knockdown cells lead to radio sensitization. Higher ROS levels correlate with increased DNA damage, and knockdown of HuR in triple-negative breast cancer (TNBC) cells results in oxidative stress and DNA damage, thereby enhancing radio sensitivity [99]. Polychlorinated biphenyls (PCBs), which are environmental pollutants that bioaccumulate in the food chain and are concentrated in adipose tissue, including breast tissue [100], can generate free radicals and oxidative DNA damage during their oxidation. This process is implicated in the development of breast cancer [101]. Scutebarbatine A (SBT-A), a diterpene alkaloid, demonstrates strong inhibitory effects on breast cancer cells by inducing DNA damage, apoptosis, and ER stress through ROS generation, while also regulating the MAPK and EGFR/Akt signaling pathways [102]. Psoralidin (PSO), a natural phenolic coumarin, induces DNA damage and protective autophagy in MCF-7 cells through ROS generation via a NOX4-dependent mechanism [103].

### 2.2. The Impact of ROS on the Breast Cancer Tumor Microenvironment

The TME of breast cancer is a complex and heterogeneous ecosystem composed mainly of tumor cells, stromal cells [104], immune cells [105], and various other components. Fibroblasts within the TME provide structural support and play a crucial role in the formation of the extracellular matrix (ECM) [106,107], influencing the growth and migration of tumor cells while facilitating intercellular signaling. Endothelial cells contribute to blood vessel formation and ensure the supply of oxygen and nutrients to tumor cells. Immune cells, including macrophages, lymphocytes, and dendritic cells, are critical for immune evasion by tumor cells, which is facilitated by multiple layers of immunosuppression in breast cancer TME. The TME is also enriched with various biological factors, such as cytokines, chemokines, and growth factors, that impact tumor growth and metastasis by regulating cellular behavior and promoting cell proliferation and migration [108]. Consequently, the TME can facilitate tumor growth through various mechanisms, including supplying oxygen and nutrients, secreting protumor factors and cytokines, and suppressing immune responses against tumor cells [109,110].

#### 2.2.1. ROS Participate in the Regulation of Tumor Immunity and Inflammation

The inflammatory TME plays a crucial role in the development and progression of tumors [111]. Tumor cells can release various inflammatory mediators, including ROS, prostaglandin E2 (PGE2), interleukins, and interferons, which collectively influence tumor development by regulating immune responses and promoting tumor cell growth [112]. ROS, in particular, can activate signaling pathways such as NF-κB and induce the production of inflammatory cytokines, thereby initiating the formation of an inflammatory TME. The accumulation of ROS can lead to oxidative stress, which may directly damage DNA or indirectly activate cell signaling pathways. This damage can induce somatic cell mutations and trigger tumor transformation while impairing the normal functions of immune cells and inflammatory cells, which are recruited to tumor sites. The presence of these cells in the TME promotes the maintenance of a chronic inflammatory environment [113]. Emerging evidence indicates that early changes in monocyte or macrophage function are associated with increased ROS production, thiol modification and disruption of redox-sensitive signaling pathways [114,115].

The immunosuppressive TME is typically characterized by moderate acidity, GSH overexpression, hypoxia, low levels of immunogenic T cell infiltration, and inefficient antigen presentation. The loss of extracellular redox regulation creates a microenvironment conducive to cancer progression [116]. Reprogramming “cold” tumors into “hot” tumors, which are more amenable to immunotherapy, may involve inducing cytotoxic ROS generation through strategies such as photodynamic therapy (PDT)/chemodynamic therapy (CDT). These approaches can stimulate an immune response, promote tumor cell apoptosis and necrosis, enhance exposure to tumor-associated antigen (TAA), and improve the infiltration of immunogenic T cells, thereby increasing antitumor efficacy [117]. Moreover, high-dose vitamin C (VitC) has been shown to inhibit PD-L1 transcription via the ROS-pSTAT3 signaling pathway, enhancing the infiltration and function of CD8^+^ T cells within the TME [118]. While early-stage solid tumors can often be effectively treated via surgical resection, the presence of residual microtumors, a proinflammatory microenvironment, and activated platelets at the surgical site can lead to tumor recurrence and metastasis, ultimately resulting in a poor prognosis. To address this issue, a ROS-scavenging gel loaded with anticancer drugs can be injected into the surgical site to enhance the TME and inhibit the recurrence of residual microtumors [119]. Research by Lu Wang et al. developed a 3D-printed prosthesis containing therapeutic hydrogel. By combining ROS-reactive hydrogels with 3D-printed breast prostheses, this innovative approach allows for personalized shape reconstruction, while simultaneously detecting and inhibiting tumor recurrence [120]. Tumor-related chronic inflammation significantly limits the efficacy of tumor immunotherapy. Xiuqi Liang et al. have developed a ROS-responsive programmable release hydrogel-based engineered scaffold (PHOENIX) to disrupt the balance of chronic inflammation in cold tumors, triggering a robust immune response [121]. In a study by Zhao et al., an anti-tumor drug designed to amplify oxidative stress, termed PBCH, was developed. This drug produces a burst of ROS in the acidic TME, leading to immunogenic cell death and effectively inhibiting growth and metastasis of TNBC [122]. Particularly relevant for basal-like breast cancer (BLBC) and *BRCA1*-associated breast cancers, studies show that ROS levels correlate with the expression and activity of the transcription factor aryl hydrocarbon receptor (AhR). Mechanistically, ROS facilitate the nuclear accumulation and activation of AhR, promoting the transcription of antioxidant enzymes and epidermal growth factor receptor (EGFR) binder-regulin (AREG). This process also stimulates chemokine production, attracting monocytes and activating the proangiogenic functions of macrophages within the TME. Importantly, the levels of these chemokines and the infiltration of monocytic lineage cells (including monocytes and macrophages) are positively correlated with ROS levels in BLBC [123].

#### 2.2.2. In the Hypoxic Tumor Microenvironment, ROS Are Involved in Inducing Metabolic Shifts in Tumor Cells

Tumor cells often face low nutrient and oxygen availability, prompting them to stimulate angiogenesis to adapt to hypoxic conditions [124,125]. Hypoxia induces the activation of transcription factors such as hypoxia-inducible factor-1α (HIF1α) and forkhead box protein o1 (FoxO1), which drive epigenetic changes that upregulate phosphoenolpyruvate carboxylase (PCK1) and facilitate the transition from gluconeogenesis to glycogen synthesis and glycolysis. This metabolic reconfiguration leads to increased NADPH production, promoting reduced GSH synthesis, slightly elevating ROS levels, and stimulating the growth of hypoxic tumor-relevant cells [126,127]. Under hypoxic conditions, elevated ROS, byproducts of rapid metabolism, can inactivate pyruvate kinase M (PKM2) by modifying critical sulfhydryl groups. This modification drives metabolic reprogramming and promotes cancer progression mediated by HIF-1α [128]. Additionally, interleukin-32 beta (IL-32β) translocates to mitochondria under hypoxic conditions and is associated with mitochondrial biogenesis, linking it to either oxidative phosphorylation (OXPHOS) or glycolysis. Specifically, IL-32β has been shown to activate lactate dehydrogenase, thereby enhancing glycolysis. This indicates that the hypoxia-ROS-IL-32β-Src-glycolysis pathway plays a significant role in regulating cancer cell metabolism [129]. BLBC is known for its aggressive nature, high metastatic potential, and resistance to chemotherapy. The silencing of fructose-1,6-bisphosphatase (FBP1) increases cancer cells’ reliance on glucose metabolism, concurrently reducing ROS levels through decreased mitochondrial respiration while enhancing NADPH production via the pentose phosphate pathway. Such lower ROS levels can promote epithelial-mesenchymal transition (EMT) and enhance cancer stem cell (CSC) phenotypes [130]. Normal mammary epithelial cells respond to transient mechanical stimuli through a calcium signaling mechanism, with persistent calcium levels maintained by microtubule-dependent mechanical activation of ROS generated by NADPH oxidase 2 (NOX2). Clinical and experimental data suggest that in breast cancer patients with KRAS activation, inhibiting mechanically induced calcium influx by ROS affects cellular responses to the tumor’s mechanical microenvironment, potentially impacting patient survival outcomes [131].

### 2.3. Role of ROS and Hormones in Breast Cancer

A family history of breast cancer is one of the most significant risk factors for developing the disease. However, a variety of non-genetic risk factors also contribute to the disease, which can be categorized broadly into hormonal and nonhormonal factors [132]. Nearly 75% of breast cancers express the ER and/or the progesterone receptors (PR), while up to 20% overexpress human epidermal growth factor receptor 2 (HER2) or exhibit HER2 amplification. Notably, about 50% of all HER2-overexpressing breast cancers also demonstrate co-expression of ER and/or PR [133]. Oxidative stress plays a critical role in the development and progression of breast cancer by modulating hormonal activity (Figure 2). Hormones can regulate the expression of genes involved in oxidative stress responses, either by directly influencing antioxidant pathways or indirectly affecting mitochondrial function.

#### 2.3.1. Regulatory Role of ROS in Estrogen-Sensitive Breast Cancer

During breast cancer development, estrogens function as ligands to activate ER in both genomic and nongenomic ways. Estrogen promotes ROS production via mitochondrial metabolism [134]. Notably, estrogen-induced ROS production and subsequent DNA strand breaks have been documented in estrogen-sensitive cells for decades [135]. Research by Shaolong Zhang and colleagues illustrates the kinetics of estrogen-induced ROS and the generation of DSB, categorizing them into two types. DSBs associated with oxidative DNA damage primarily occur in the most fragile and active chromatin regions. Other DSBs arise from the activities of enzymes such as apolipoprotein b mRNA-editing enzyme-catalytic polypeptide-like 3b(APOBEC3B) and topoisomerase I/II, which facilitate chromatin remodeling and promote estrogen signaling by relieving torsional stress [136]. Interestingly, limited ROS production can sufficiently reduce DNA damage and delay tumorigenesis in a *BRCA1*-deficient breast cancer mouse model. In this model, large amounts of endogenous estrogen oxidative metabolites induce DNA adducts and create apyrimidinic/apyrimidine sites linked to DSBs and genomic instability in the mammary gland. Antioxidant therapy that inhibits estrogen oxidation has been shown to reduce oxidative DNA damage and postpone the onset of breast tumors [137]. Furthermore, knocking out ERα can trigger autophagy and inhibit the activation of unfolded protein response (UPR) mediated by antiestrogen treatments, enhancing ROS-induced cell death in breast cancer [138,139]. Hours after E2 stimulation in endocrine-resistant breast cancer cells, the UPR is initiated, particularly via activation of the PRK-like endoplasmic reticulum kinase (PERK). PERK regulates the production of metabolites such as lipids, ROS, and Ca^2+^, thereby influencing E2-induced apoptosis [140].

#### 2.3.2. The Effects of Various Hormones on Breast Cancer Related to ROS

Beyond estrogen, various hormones significantly impact the growth and metastasis of breast cancer. For instance, PRs have been shown to mediate STAT signaling pathways associated with inflammation and immunity, regulating breast cancer cell responses [141]. Glucocorticoid receptors also play a crucial role in mediating oxidative stress and inflammation, with evidence indicating that these receptors are highly active in breast cancer and metastatic sites. Synthetic glucocorticoid derivatives are commonly used clinically as anti-inflammatory drugs and immunosuppressant agents [142]. The role of androgens and androgen receptors (ARs) in breast cancer remains somewhat controversial; however, studies by some scholars have shown that they have been linked to enhanced cancer invasion and metastasis [143]. Additionally, the cyclic neuropeptide oxytocin (OT) initiates various biological responses in both the central and peripheral nervous systems via the oxytocin receptor (OTR). These responses include social bonding, oxidative stress, maternal behavior, sexual activity, uterine contractions, milk excretion, and implications for cancer development. Growing evidence indicates that OTR is involved in breast cancer development and progression, with multiple breast cancer cell lines expressing this receptor [144]. Recent studies have also identified associations between other hormones and an increased risk of breast cancer, including the effects of thyroid hormones [145,146,147] and obesity [148,149]. Such findings highlight the complex interplay between hormonal regulation, ROS production, and breast cancer dynamics.

#### 2.3.3. Hormone Abnormalities Caused by Obesity and Their Link to Breast Cancer Progression

The adipose tissue microenvironment plays a crucial role in the occurrence and development of breast cancer. Specifically, adipokines released by adipocytes function as soluble factors with hormone-like functions, influencing tumor development and exhibiting both pro-inflammatory and anti-inflammatory effects [150]. Key adipokines include leptin, adiponectin, and inflammatory cytokines such as tumor necrosis factor-alpha (TNF-α) and interleukin-6 (IL-6) [151]. Various signaling products from the transcription factor nuclear factor κB, particularly inflammatory eicosanoids, ROS, and cytokines, are implicated in cancer associated with chronic inflammation [152,153]. In breast cancer, locally active eicosanoids and adipokines may play critical roles within the TME. Several growth factors have been identified as stimulators of cyclooxygenase-2(COX-2) activation, which can lead to secondary inflammatory responses [154].

Obesity is linked to elevated levels of various hormones, including insulin, which increases the risk of breast cancer and is associated with more aggressive cancer phenotypes. Insulin signaling has been shown to stimulate breast cancer cell proliferation and survival, highlighting the importance of metabolic health in managing breast cancer risk. The growth hormone-insulin-like growth factor 1 (GH-IGF-1) axis also significantly impacts breast development and tumorigenesis, interacting with estrogen signaling pathways to influence both normal and malignant processes. High expression of insulin-like growth factor type 1 receptor (IGF-1R) is associated with hormone receptor-positive breast cancer. Inhibition of IGF-1R in mouse or human tumor epithelial cells has been shown to increase ROS production and activate the endoplasmic reticulum stress response. IGF-1R signaling in breast tumor epithelial cells protects against endoplasmic reticulum stress and modulates the TME [155]. Epidemiological studies have linked hyperinsulinemia and type 2 diabetes to an increased risk of breast cancer, as well as aggressive and metastatic phenotypes, and poor prognosis. Insulin stimulates the proliferation of certain human breast cancer cell lines in vitro through the phosphatidylinositol-3 kinase and mitogen-activated protein kinase (MAPK)/Akt signaling pathways. Additionally, insulin acts as an anti-apoptotic agent, enhancing the migration and invasion capabilities of tumor cells [156]. Liraglutide, an insulin secretagogue and glucagon-like peptide-1 (GLP-1) analog, activates the glucagon-like peptide-1 receptor (GLP-1R) by enhancing insulin secretion from pancreatic β cells. This process promotes cell proliferation and migration through the NOX4/ROS/VEGF signaling pathway, playing a significant role in the progression of breast cancer [157].

## 3. Dilemmas and Prospects of Breast Cancer Treatment

### 3.1. Relationship Between ROS and Drug Resistance in Breast Cancer

Drug resistance presents a significant challenge in cancer therapy, particularly in breast cancer management. Tumor cells can develop resistance to drugs over time, a phenomenon known as acquired resistance. This acquired drug resistance is often linked to modifications in the TME, alterations in intracellular metabolisms, and changes in gene expression [158,159]. In breast cancer research, ROS are recognized as crucial factors influencing tumor progression and drug resistance. ROS can promote drug resistance through several mechanisms, including increased drug efflux, alterations in target genes, inactivation of drugs within cells, and activation of signaling pathways that lead to epithelial-mesenchymal transition (EMT) and DNA repair mechanisms [160,161].

#### 3.1.1. ROS and Resistance to Radiotherapy and Chemotherapy in Breast Cancer

Resistance to DNA-damaging agents is a major contributor to treatment failure in tumors. Chemotherapeutic agents, such as cisplatin and radiotherapy are essential components of cancer treatment; however, the emergence of resistance poses a significant obstacle, often resulting in tumor recurrence and increased mortality [162,163,164]. Elevated ROS levels can modify the targets of certain drugs, hindering their ability to bind effectively and thus reducing their therapeutic efficacy [160,165]. Cancer cells often adapt to severe circulatory hypoxia and intermittent reoxygenation, leading to an increased cellular antioxidant capacity that supports resistance to chemotherapy and radiotherapy. This antioxidant system includes antioxidant enzymes: superoxide dismutase (SOD), catalase (CAT), and GSH-Px, as well as GSH, vitamin C and vitamin E, and carotenoids [166]. In response to elevated ROS levels during treatment, cancer cells frequently upregulate antioxidant enzymes like SOD and GSH, enhancing their ability to withstand oxidative stress and increasing resistance to chemotherapeutic agents [167,168].

Key regulators of redox signaling, including phosphatases and kinases, are influenced by oxidants, propagating signals that lead to various biological responses. These signaling pathways can significantly affect the effectiveness of chemotherapy and radiotherapy [169,170]. Drug resistance mediated by ROS may also arise from the activation of redox-sensitive transcription factors, such as nuclear factor-kappa B(NF-κB), nuclear factor-erythroid 2-related factor 2 (Nrf2) [171], cellular-jun (c-Jun) [172], and HIF-1α [173]. Additionally, in cells with mitochondrial dysfunction, increased expression of NADPH oxidase (NOX) drives tumor metabolism toward glycolysis [174]. This metabolic change is facilitated by the upregulation of DNA repair mechanisms, enhanced pro-survival signaling, activation of autophagy, and the induction of drug efflux pumps, all of which contribute to ROS neutralization and promote cancer cell proliferation and survival [175]. Therapeutic targets for cancer stem cells (CSCs) in TNBC include NAD(P)H quinone oxidoreductase 1 (NQO1) and superoxide dismutase 1 (SOD1). Targeting these enzymes can eliminate CSCs by increasing mitochondrial oxidative damage and activating apoptotic pathways, presenting a potential application in overcoming treatment resistance in TNBC [176]. Furthermore, interference in mitochondrial glutathione (mGSH) levels is associated with overcoming apoptotic resistance in cancer cells. Inhibiting the GSH maintenance transport system disrupts cellular and mitochondrial redox homeostasis, thereby sensitizing cancer cells to pro-oxidant treatments such as radiotherapy. The GSH maintenance transport system includes glutamate oxaloacetate aminotransferase 1(GOT1) [177], mitochondrial citrate carrier SLC25A1 [178], mitochondrial dicarboxylate carrier (SLC25A10), and mitochondrial 2-oxoglutarate carrier (SLC25A11) [179].

#### 3.1.2. ROS Are Associated with the Development of Resistance to Endocrine Therapy in Breast Cancer

In breast cancer, estrogen induces tumor cell proliferation and promotes disease progression [180]. Endocrine therapy drugs, such as tamoxifen (TAM) and other antiestrogens, as well as the aromatase inhibitor exemestane, can generate ROS. Invasive breast cancer cells have high oxidative stress, and long-term treatment with exemestane, fulvestrant, or TAM may increase additional ROS stress. Breast cancer cells receiving long-term antiestrogen therapy appear to adapt to this increased and sustained level of ROS. Resistance to endocrine therapy may be mediated in part through ROS-mediated dysregulation of estrogen-dependent and estrogen-independent red oxygen reduction-sensitive signaling pathways [181].

After extensive clinical trials, in 1977, the U.S. Food and Drug Administration (FDA) approved TAM, an estrogen inhibitor for the treatment of breast cancer [182]. Studies have demonstrated that mitochondrial function decreases and that the mitochondrial network becomes fragmented in TAM-resistant cells. This compromised mitochondrial function, combined with elevated ROS levels and metabolic adaptation, allows tumor cells to develop resistance to TAM [180,183]. One study investigating the effect of TAM on breast cancer cell activity revealed a correlation with the release of mitochondrial cytochrome c, a decrease in the mitochondrial membrane potential, and an increase in ROS levels. At pharmacological concentrations, TAM rapidly induces mitochondrial changes within breast cancer cells, potentially triggering apoptotic pathways in the tumor tissue of TAM-treated patients. These mechanisms could explain the capacity of TAMs to promote apoptosis in breast cancer cells [184,185], suggesting that increased ROS levels, which inhibit mitochondrial function, are crucial for enhancing the efficacy of TAM therapy.

Glutathione S-transferase Mu 3 (GSTM3) is an enzyme involved in the detoxification of electrophilic compounds by coupling to GSH. GSTM3 mRNA is highly expressed in ER- and HER2-positive breast cancer. MCF-7 cells silenced by GSTM3 are more sensitive to H2O2 and have the ability to significantly inhibit proliferation and colony formation. Tamoxifen-resistant (Tam-R) cells lacking GSTM3 have increased sensitivity to H_2_O_2_, and GSTM3 silencing may inhibit the tumorigenic ability of MCF-7 cells and increase tumor cell apoptosis [186]. Ferroptosis is a non-apoptotic form of cell death driven by iron-dependent lipid peroxidation and is associated with perturbations of redox homeostasis [187,188]. Some studies have indicated that ROS induced by TAM enhances ferroptosis in breast cancer cells, whereas the activation of glutathione peroxidase 4(GPX4) inhibits this process. Resensitization to TAM therapy also remains an important consideration in treatment strategies [189]. Research has demonstrated that GATA3 promotes cell viability by downregulating the expression of the ferroptosis-related gene CYB5R2 and maintaining iron homeostasis, thereby facilitating the development of drug resistance in luminal breast cancer cells [190].

### 3.2. Breast Cancer Treatment Drugs and Methods That Modulate Oxidative Stress

Understanding the balance between ROS production and antioxidant defense is crucial for developing effective breast cancer therapies. Strategies that increase ROS production in cancer cells or inhibit their antioxidant responses could significantly improve treatment outcomes. Moreover, the timing and combination of therapies that modulate oxidative stress may be key to overcoming drug resistance and improving patient survival [191]. Consequently, breast cancer treatments that utilize oxidative stress primarily include chemotherapy drugs and radiation therapy. Below is a detailed overview of these treatments and their mechanisms of action.

#### 3.2.1. Breast Cancer Treatment Drugs Rely on ROS to Play a Major Role

Many traditional chemotherapeutic agents induce oxidative stress in cancer cells as a mechanism for triggering cell death. Drugs commonly used in chemotherapy regimens for breast cancer include alkylating agents (e.g., cyclophosphamide); anthracyclines (e.g., doxorubicin and epirubicin); antimetabolites (e.g., fluorouracil and methotrexate); plant derivatives (e.g., paclitaxel and polypeptide docetaxel); and platinum-based compounds. Numerous chemotherapy agents generate ROS, leading to oxidative damage to cancer cells [192,193]. Since the antioxidant defense systems in cancer cells are often compromised, these cells struggle to cope with the oxidative stress imposed by these medications. Some targeted therapies aim to intentionally increase oxidative stress in cancer cells. For example, certain drugs can inhibit the antioxidant defenses of cancer cells, thereby increasing their sensitivity to oxidative damage. This approach often involves targeting the GSH system, which is essential for maintaining redox balance within cells. By depleting GSH levels, these therapies can increase ROS levels, resulting in increased oxidative stress and subsequent death in cancer cells [194].

Studies have shown that silencing the *SIRT3* gene, which plays a pivotal role in regulating mitochondrial ROS production, can increase the sensitivity of breast cancer cells to cisplatin and TAM through increased oxidative stress and the promotion of apoptosis [195]. Furthermore, research indicates that targeting BTB and CNC Homology 1(BACH1), a transcription factor involved in oxidative stress responses, can sensitize breast cancer and potentially other tumor types to mitochondrial inhibitors [196]. Another mechanism involves gasdermin, which triggers doxorubicin (DOX)-induced pyroptosis through a caspase-3-dependent reaction mediated by the ROS/JNK signaling pathway. Reducing the protein level of GSDME may diminish DOX-induced toxicity and pyroptosis in breast cancer cells [197]. Additionally, interest in the use of dietary antioxidants and phytochemicals to modulate oxidative stress in cancer therapy is increasing. Compounds such as curcumin [198], resveratrol [199], and sulforaphane [200] have been investigated for their potential to selectively increase oxidative stress in cancer cells while sparing normal cells. These natural compounds can influence various signaling pathways associated with oxidative stress responses, thereby promoting cancer cell apoptosis [201,202]. Certain natural compounds, such as flavonoids, have been identified as having the potential to modulate oxidative stress in cancer therapy. These compounds can affect various signaling pathways involved in carcinogenesis and may enhance the effects of conventional therapies by increasing oxidative stress in cancer cells [203].

#### 3.2.2. Innovative Breast Cancer Treatment Combined with ROS

Although chemotherapy remains a mainstay in cancer treatment, reducing side effects and maximizing treatment effectiveness continue to pose significant challenges. Prodrug design offers a solution by inactivating potent but nonselective drugs and reactivating them only at the tumor site through stimulus–response mechanisms. Controlled release of prodrugs can be facilitated via internal or external stimuli such as pH, enzymes, ROS, magnetic fields, light, and ultrasound [204]. Owing to both the efficacy and side effects associated with doxorubicin (Dox) prodrugs, significant efforts have been made to develop variants, including boron- or sulfur-based ROS linkers [205,206]. Mina Jafari, for example, successfully designed and synthesized a novel set of modified peroxy-amide-based doxorubicin ROS-reactive prodrugs [207]. Innovative delivery systems also show promise in enhancing existing treatments by targeting oxidative stress pathways; for example, hyperthermia combined with nanoparticle-mediated drug delivery has demonstrated potential in inducing oxidative stress in cancer cells. Evidence indicates that lanthanum strontium manganese oxide nanoparticles can induce apoptosis and autophagy in breast cancer cells by increasing oxidative stress [203]. Some experimental treatments focus on the direct induction of ROS in cancer cells. For example, long-term exposure to hydrogen peroxide has been shown to promote the growth and survival of breast cancer cells, suggesting that controlling ROS levels can influence tumorigenesis [202].

Interestingly, artificial nanozyme-mediated tumor therapy has attracted significant attention because of its ability to actively leverage the characteristics of the TME. This approach has advantages such as low cost, controllable catalytic activity, and high stability [208]. Various nanozymes, including alloy nanoparticles [209], metal chalcogenides [210], and single-atom catalysts [211], have been designed and engineered to generate ROS for tumor destruction. In addition, novel therapeutic strategies that synergistically combine cascade enzymes, electrodynamics, ferroptosis, and immunotherapy within a single nanomedicine framework are emerging. One promising development is the synthesis of a bayberry-shaped PtMnIr nanozyme, which has multiple enzymatic activities, including oxidase (OXD), catalase (CAT), SOD, peroxidase (POD), and GPX activities. This nanozyme continuously generates ROS while consuming GSH in an “internal catalytic loop”, ultimately enhancing tumor treatment efficacy [212]. Moreover, drug development efforts focused on ROS-assisted therapy aim to identify and selectively modulate specific ROS-producing enzyme sources involved in disease pathology, ensuring that the physiological signaling and metabolic functions of ROS remain intact. Among these enzyme sources, the NADPH oxidase (NOX) family is uniquely responsible for ROS formation. Advances in the selective targeting of disease-related, ROS-related proteins have led to multiple phase II/phase III clinical trials, alongside the introduction of several registered drugs [213]. One innovative example includes the development of a bioactive, CD44-targeting hyaluronic acid nanoparticle that encapsulates the NOX inhibitor GKT831 (HANP/GKT831). Systemic delivery of HANP/GKT831 facilitates targeted accumulation in patient-derived xenograft (PDX) tumors in nude mice. This targeted delivery subjects tumor cells to radiation-induced DNA damage and apoptosis by downregulating DNA repair mechanisms and oncogenic signaling pathways [214].

Photodynamic therapy (PDT) has also emerged as a widely used approach for antitumor treatment, drawing interest for its noninvasive nature, minimal side effects, and low potential for drug resistance [215,216]. Zheng Huang’s development of a photosensitizer prodrug (CSP) enhances the efficacy of PDT by increasing intracellular ROS levels while simultaneously reducing GSH levels, which decreases ROS consumption. This mechanism facilitates the production of a substantial amount of highly cytotoxic ROS, thereby synergistically amplifying DNA damage and impairing DNA repair mechanisms. Consequently, these advancements improve therapeutic effectiveness against TNBC cells [217]. Overall, the integration of innovative nanozyme technologies and targeted therapies leveraging ROS highlight a promising frontier in cancer treatment. By optimizing ROS generation and modulation within the TME, these strategies may significantly improve therapeutic outcomes and overcome challenges associated with conventional therapies. Continued research and development in this area are essential for translating these approaches into effective clinical applications for cancer patients.

In summary, manipulating ROS levels presents opportunities to enhance treatment efficacy in breast cancer, particularly in overcoming drug resistance. Understanding the interplay between ROS dynamics, mitochondrial function, and cell survival pathways can pave the way for novel therapeutic approaches that improve patient outcomes in breast cancer treatment. Future research should focus on developing combination strategies that target both ROS modulation and metabolic pathways, potentially leading to more effective interventions for resistant breast cancer subtypes.

#### 3.2.3. Breast Cancer Treatment Combined with ROS-Mediated Cell Death Regulation

Regulated Cell Death (RCD) is a precisely controlled cell death process that plays a critical role in various physiological and pathological processes, including cellular homeostasis, embryonic development, tissue repair, and immune defense. RCD encompasses several forms of cell death, including apoptosis, necroptosis, ferroptosis, pyroptosis, PANoptosis, etc. [218]. Exploiting the role of ROS in these cell death pathways may offer new targets and strategies for breast cancer treatment. By modulating ROS levels to influence specific cell death mechanisms, it may be possible to induce tumor cell death and achieve therapeutic outcomes in breast cancer.

Notably, breast cancer stem cells (BCSCs) exhibit low oxidative stress to maintain their viability and plasticity. The histone reader zinc finger mynd-type containing 8(ZMYND8) is upregulated in BCSCs and interacts with nuclear factor-erythroid 2-related factor 2(NRF2), recruiting it to the promoters of antioxidant genes. This interaction enhances gene transcription in mammospheres, thus promoting BCSC stemness and tumorigenesis by inhibiting ROS production and ferroptosis [219]. Compounds from the β-nitrosstyrene family, originally identified as slow-binding inhibitors of protein tyrosine phosphatase, have emerged as potential therapeutic agents. The synthetic derivative CYT-Rx20 induces programmed cell death in breast cancer cells [220]. By inhibiting the TNFα/NFκB signaling pathway in a retinoid X receptor α (RXRα)-dependent manner, resulting in apoptosis [221]. Tumor immunogenic cell death (ICD) generates neoantigens when tumor cells undergo cytotoxicity due to external stimuli. ICD is a unique form of apoptosis characterized by an apoptotic cell’s ability to provide an adaptive immune response. This approach can transform a non-immunogenic environment into an immunogenic one, facilitating anti-tumor immune responses [222]. For instance, Ophiosin, found in marine fungi or sponges, can induce immunogenic cell death against triple-negative breast cancer via the ER stress-CHOP pathway, demonstrating effective anti-tumor activity [223]. In 1999, Chi et al. discovered that Ras could trigger glioblastoma cell death featuring cytoplasmic vacuoles, which differed from classic apoptosis yet aligned with autophagy’s morphological characteristics. This cell death process was termed ‘methuosis’ [224]. A novel methuosis inducer, DZ-514, exhibits anti-tumor activity in triple-negative breast cancer by activating the ROS-MKK4-p38 axis [225]. Natural compounds like Arabinogalactan (AG) and curcumin (Cur) have also been extensively studied for their potential in cancer therapy. The combination of AG and Cur has been shown to promote breast cancer cell apoptosis in breast cancer cells by elevating ROS levels, altering mitochondrial membranes, and reducing glutathione levels [226]. In summary, a deeper understanding of the role of ROS in regulated cell death patterns in breast cancer is crucial for unraveling the pathological mechanisms underlying this disease and developing new therapeutic strategies.

## 4. Discussion

In this review, we summarized the role of ROS in affecting the occurrence and development of breast cancer, as shown in Table 1. Drugs that modulate oxidative stress exhibit preferential anticancer effects by influencing various cellular processes, including apoptosis, DNA damage, ER stress, autophagy, metabolism, and migration. A significant challenge in cancer research is achieving selective effects on cancer cells while minimizing harm to normal cells [227]. Understanding the balance between oxidants and antioxidants in breast cancer is crucial for developing effective treatment strategies. Recent studies have shown that breast cancer risk factors are closely related to oxidative stress, where both oxidants and antioxidants play distinct roles. The potential of combining different antioxidants to enhance therapeutic outcomes in breast cancer patients has demonstrated promising results [228].

Here, we highlight the central role of oxidative stress and ROS-regulated functions in breast cancer therapy. The varied levels of ROS induced by oxidative stress-regulating drugs influence critical processes, including apoptosis, DNA damage, ER stress, autophagy, metabolism, and migration, and underscore their interactions. Given the current research progress on oxidative stress in breast cancer, we propose several avenues for future therapeutic developments:(1)Development of drugs targeting oxidative stress: Breast cancer cells typically exhibit elevated oxidative stress levels, which they exploit for survival and proliferation. Therefore, drugs targeting these redox pathways may effectively inhibit tumor growth. Creating oxidative stress inducers that selectively target and kill tumor cells represents a potentially transformative therapeutic strategy. For instance, natural plant extracts or derivatives such as picartinol, resveratrol glycosides, pterostilbene, and ursolic acid can regulate intracellular redox status and influence cancer cell proliferation, apoptosis, migration, and invasion [10].(2)Application of antioxidants: While antioxidants are primarily employed to protect normal cells from oxidative stress, they can also increase the sensitivity of breast cancer cells when used alongside chemotherapy agents. However, this combination should be approached with caution, as antioxidants can occasionally interfere with the therapeutic effects of these treatments.(3)Development of biomarkers: Biomarkers associated with oxidative stress may play an essential role in assessing the prognosis and treatment response of patients with breast cancer. For example, measuring oxidative stress markers in tumor tissue or blood can inform personalized treatment approaches. However, there are limitations to this method: Many oxidative stress markers are not specific to tumors and may also change in other pathological conditions such as inflammation and infection. Additionally, the sample collection, processing, and storage process significantly impact test results. Improper handling, such as prolonged exposure of tissue samples to room temperature or contamination of blood samples during centrifugation, can alter oxidative stress marker content and affect test accuracy.(4)Combination with immunotherapy, oxidative stress is intricately connected to the immune response. Current studies are exploring how to leverage the antitumor immune response induced by oxidative stress in conjunction with immunotherapy to enhance therapeutic efficacy. For instance, ROS-generating drugs and immuno-therapeutics can be co-encapsulated in nanoparticles for co-delivery [212]. Combining ROS with immune checkpoint inhibitors such as anti-PD-1/PD-L1 antibodies may enhance T cell-mediated killing of breast cancer cells [229,230]. Since the concentration and action time of drugs in the body are difficult to predict, which will affect the effect of combination therapy and even increase the risk of adverse drug reactions, the difficulty that must be overcome when researching and using related ROS–immune combination therapy is to improve tumor targeting and the dose effect to reduce the occurrence of toxic side effects.(5)Combining hormone therapy with other targeted therapies is a promising direction, so combining endocrine therapy drugs with drugs that can induce ROS production and detecting indicators such as hormone receptor status and intracellular ROS levels in the tumor tissues of breast cancer patients can lead to the development of more personalized treatment plans for patients. For patients whose treatment is ineffective or resistant to hormone therapy, combined ROS modulation therapy may be a new treatment strategy. Based on the concept of combined hormone and ROS therapy, new multifunctional drugs can be developed. This drug can regulate both hormone signaling and ROS levels.

In summary, oxidative stress holds great promise in the treatment of breast cancer, yet further research is needed to elucidate its mechanism and optimal applications. Future treatment strategies should focus on rationally regulating oxidative stress levels not only to increase the sensitivity of tumor cells to therapies but also to safeguard the health of normal cells. By continuing to explore the intricate relationship between oxidative stress and cancer pathophysiology, we can improve therapeutic outcomes and patient quality of life.

## Figures and Tables

**Figure 1 antioxidants-14-00104-f001:**
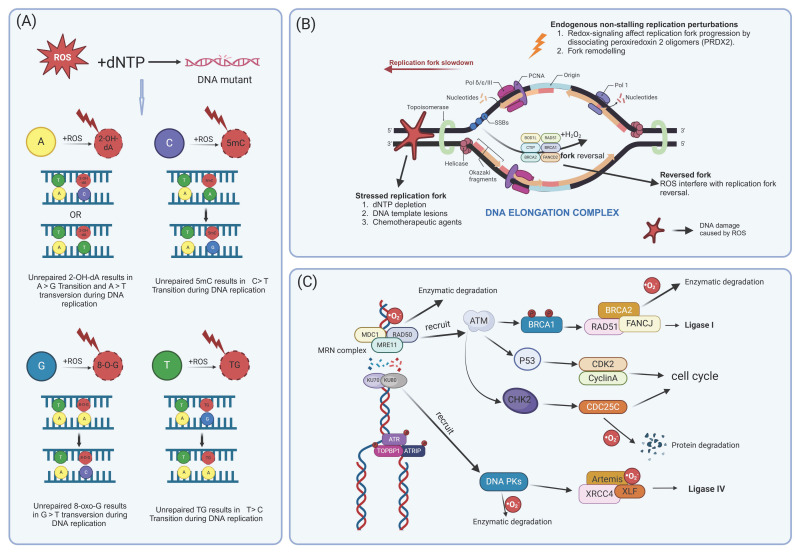
Excessive ROS lead to DNA damage. (**A**) ROS contribute to the oxidation of dNTP, leading to DNA damage; (**B**) oxidative stress disrupts replication fork function and slows down replication fork progression. (**C**) The function of intracellular DNA repair enzymes is compromised due to ROS oxidation, resulting in impaired DDR damage response. (Created with BioRender.com).

**Figure 2 antioxidants-14-00104-f002:**
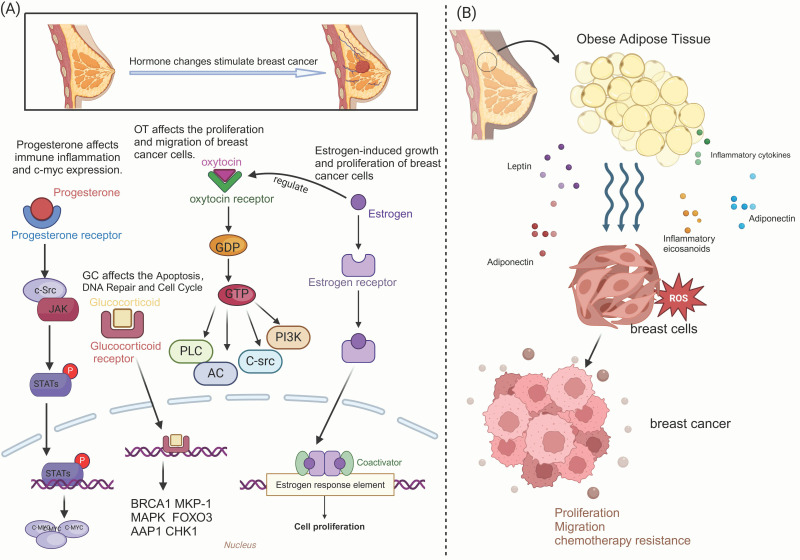
Role of ROS and hormones in breast cancer. (**A**) The regulatory role of various hormones in breast cancer development. (**B**) The adipokine secreted by the surrounding adipose tissue regulates the occurrence and progression of breast cancer and is related to ROS. (Created with BioRender.com).

**Table 1 antioxidants-14-00104-t001:** Impact of ROS on breast cancer progression.

Oxidative stress causes breast cancer development.	Cause DNA damage and repair abnormalities.	dNTPs are easily oxidized.	Traditional chemotherapy drugs induce oxidative stress, such as alkylation, anthracyclines and other drugs; the development of drugs targeting oxidative stress, such as targeting the GSH system, SIRT3, etc.
		Replication stress occurs, leading to genetic mutations.	
		Affects the activity and function of repair enzymes.	
	Change the tumor microenvironment.	Immune inflammatory environment.	Prodrug design is combined with nanotechnology, such as ROS-responsive prodrugs, nanozymes combined with immunotherapy to regulate ROS and improve T cell function; develop antioxidant enzymes and oxidation product markers, such as SOD, MDA, etc.
		Induces metabolic shifts in tumor cells.	
		Extratumoral cell matrix changes.	
	Regulation of hormones affecting breast cancer.	Estrogen-sensitive breast cancer.	Combination of photodynamics, hormones and ROS regulation; combination of ROS and hormone-targeting drugs, such as AR, ER antagonists, and ROS inducers.
		PR; AR; OTR	
		Hormone abnormalities associated with obesity.	

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
