# Peer review of "Role of Oxidative Stress in the Occurrence, Development, and Treatment of Breast Cancer"

_antioxidants, 2025, doi:10.3390/antiox14010104_

Round 1
Reviewer 1 Report
The article discusses how the imbalance of oxidative stress, represented as the unevenness between ROS and antioxidants, may be implicated in the initiation and progression of breast cancer. It talks about how ROS affects DNA, the tumor environment, and resistance to treatments. It further highlights the fact that management of oxidative stress could be one of the hopeful areas for new treatment strategies for cancers not responding to current treatment options.
In my opinion, this is an important article, since it contributes to the understanding of the relationship between oxidative stress and breast cancer. Such understanding may open new horizons in the development of drugs targeting the regulation of ROS levels, therefore improving patients' prognosis, overcoming therapeutic resistance, and promoting personalized and more effective therapies.
While it is well-written overall, there are a few key issues that I believe would make the article even better if discussed.
1. Abstract: The abstract is long and has details that could be left out to make it shorter. It might be better to lessen the information on how ROS works to focus more on the main idea, which is the importance of oxidative stress as a target for new treatments.
2. Introduction: The introduction repeats how crucial oxidative stress is, but it does not connect the dots very well to breast cancer. It would also help to better organize the information by connecting the existent research to the main points of the article. It would be easier to explain why oxidative stress is an important mechanism compared to others.
3. ROS and their role in breast cancer: There is a lot of data on ROS, but it is not always well linked together. It may be worth explaining the basic role of ROS in cells before linking this role to cancer development.
4. Tumor Microenvironment: Some of the information repeated from the previous section in regard to how ROS interacts with the immune system, grouping all this information under a single heading with several more examples of microenvironment-specific therapies, such as ROS-scavenging gels, would strengthen it considerably.
5. Resistance to therapies: There is some interesting information, though a few points from the previous sections repeat again in this one. Again, a shorter analysis and link of how resistance goes with oxidative stress and a few more examples of clinical trial evidence would make it stronger.
6. Conclusions and perspectives: The conclusions are wide and would be better with more specific steps for future research. This section could have benefited from suggestions such as investigating drugs that target ROS or developing combination therapies using ROS inducers along with immunotherapy.
The article discusses how the imbalance of oxidative stress, represented as the unevenness between ROS and antioxidants, may be implicated in the initiation and progression of breast cancer. It talks about how ROS affects DNA, the tumor environment, and resistance to treatments. It further highlights the fact that management of oxidative stress could be one of the hopeful areas for new treatment strategies for cancers not responding to current treatment options.
In my opinion, this is an important article, since it contributes to the understanding of the relationship between oxidative stress and breast cancer. Such understanding may open new horizons in the development of drugs targeting the regulation of ROS levels, therefore improving patients' prognosis, overcoming therapeutic resistance, and promoting personalized and more effective therapies.
While it is well-written overall, there are a few key issues that I believe would make the article even better if discussed.
1. Abstract: The abstract is long and has details that could be left out to make it shorter. It might be better to lessen the information on how ROS works to focus more on the main idea, which is the importance of oxidative stress as a target for new treatments.
2. Introduction: The introduction repeats how crucial oxidative stress is, but it does not connect the dots very well to breast cancer. It would also help to better organize the information by connecting the existent research to the main points of the article. It would be easier to explain why oxidative stress is an important mechanism compared to others.
3. ROS and their role in breast cancer: There is a lot of data on ROS, but it is not always well linked together. It may be worth explaining the basic role of ROS in cells before linking this role to cancer development.
4. Tumor Microenvironment: Some of the information repeated from the previous section in regard to how ROS interacts with the immune system, grouping all this information under a single heading with several more examples of microenvironment-specific therapies, such as ROS-scavenging gels, would strengthen it considerably.
5. Resistance to therapies: There is some interesting information, though a few points from the previous sections repeat again in this one. Again, a shorter analysis and link of how resistance goes with oxidative stress and a few more examples of clinical trial evidence would make it stronger.
6. Conclusions and perspectives: The conclusions are wide and would be better with more specific steps for future research. This section could have benefited from suggestions such as investigating drugs that target ROS or developing combination therapies using ROS inducers along with immunotherapy.
Author Response
Reviewer 1 comments:
The article discusses how the imbalance of oxidative stress, represented as the unevenness between ROS and antioxidants, may be implicated in the initiation and progression of breast cancer. It talks about how ROS affects DNA, the tumor environment, and resistance to treatments. It further highlights the fact that management of oxidative stress could be one of the hopeful areas for new treatment strategies for cancers not responding to current treatment options.
In my opinion, this is an important article, since it contributes to the understanding of the relationship between oxidative stress and breast cancer. Such understanding may open new horizons in the development of drugs targeting the regulation of ROS levels, therefore improving patients' prognosis, overcoming therapeutic resistance, and promoting personalized and more effective therapies.
While it is well-written overall, there are a few key issues that I believe would make the article even better if discussed.
Comment 1:
Abstract: The abstract is long and has details that could be left out to make it shorter. It might be better to lessen the information on how ROS works to focus more on the main idea, which is the importance of oxidative stress as a target for new treatments.
Response 1:
We appreciate your feedback on the abstract's length and detail. We have substantially revised the abstract to improve clarity and conciseness. The focus is now more sharply on the importance of oxidative stress as a target for novel treatments, with the description of ROS mechanisms significantly shortened and integrated more effectively into the overall narrative. Please see the revised abstract for the changes.
Comment 2:
Introduction: The introduction repeats how crucial oxidative stress is, but it does not connect the dots very well to breast cancer. It would also help to better organize the information by connecting the existent research to the main points of the article. It would be easier to explain why oxidative stress is an important mechanism compared to others.
Response 2:
Thank you for your feedback on the introduction, we have revised section 1.1 to strengthen the connection between ROS and breast cancer.
Comment 3:
ROS and their role in breast cancer: There is a lot of data on ROS, but it is not always well linked together. It may be worth explaining the basic role of ROS in cells before linking this role to cancer development.
Response 3:
Thank you for your feedback, we have additionally elaborated on the physiological role of ROS in cells in Section 1.2. We have reorganized and rewritten the section on ROS and their role in breast cancer to improve the flow of information
Comment 4:
Tumor Microenvironment: Some of the information repeated from the previous section in regard to how ROS interacts with the immune system, grouping all this information under a single heading with several more examples of microenvironment-specific therapies, such as ROS-scavenging gels, would strengthen it considerably.
Response 4:
Thank you for your suggestion. We have revised the Tumor Microenvironment section to consolidate the information on ROS interaction with the immune system and added several examples of micro-environment specific therapies, including ROS-scavenging gels, to strengthen this section.
Comment 5:
Resistance to therapies: There is some interesting information, though a few points from the previous sections repeat again in this one. Again, a shorter analysis and link of how resistance goes with oxidative stress and a few more examples of clinical trial evidence would make it stronger.
Response 5:
Thank you for your feedback. We have revised the section on resistance to therapies, streamlining the discussion and strengthening the link between resistance mechanisms and oxidative stress. Additional examples of clinical trial evidence have been included.
Comment 6:
Conclusions and perspectives: The conclusions are wide and would be better with more specific steps for future research. This section could have benefited from suggestions such as investigating drugs that target ROS or developing combination therapies using ROS inducers along with immunotherapy.
Response 6:
Thank you for your valuable feedback. We have revised the conclusions to be more focused and specific. A summary table has been added to highlight key findings and future research directions. In addition, the prospects of ROS in breast cancer treatment were explained in detail.

Reviewer 2 Report
The article titled "Role of Oxidative Stress in the Occurrence, Development, and Treatment of Breast Cancer" by Rui Dong et al. offers valuable insights; however, there are several areas that require attention:
- The authors should ensure that the abstract of the study includes the aim, methods, results, and conclusion, following the author instructions provided by Antioxidants.
- In the introduction, authors must provide the databases or online sources from which they obtained information for the manuscript. Additionally, they may include details on specific search terms, inclusion/exclusion criteria, data extraction, or methods employed to select relevant studies.
- The authors can elaborate on ROS actions on individual breast phenotypes and the underlying mechanisms, including signaling pathways or molecular interactions involved, to enhance the depth of understanding for readers.
- Authors could create a comprehensive table presenting the validity of the occurrence, development, and treatment of ROS in breast cancer, encompassing in vitro and in vivo models, drugs used, routes of administration, mechanisms, and outcomes of the studies.
- Authors may include a list of clinical studies related to oxidative stress employed by drugs or other sources that may be used in the development and treatment of breast cancer.
- Oxidative stress has detrimental effects on normal cells. Hence, authors may discuss the balance between the benefits of inhibiting cancer cell growth and the need to minimize harm to surrounding healthy tissues.
- The authors can include future research directions and potential study limitations of the present study.
The authors are encouraged to provide comprehensive tables utilizing literature from both in vitro and in vivo studies to validate their current research. This approach can enhance the credibility and robustness of their study findings
Author Response
Comment 1:
The authors should ensure that the abstract of the study includes the aim, methods, results, and conclusion, following the author instructions provided by Antioxidants.
Response 1:
Thank you for reminding us, we have modified the title style of this article to comply with the journal’s requirements.
Comment 2:
In the introduction, authors must provide the databases or online sources from which they obtained information for the manuscript. Additionally, they may include details on specific search terms, inclusion/exclusion criteria, data extraction, or methods employed to select relevant studies.
Response 2:
Thank you for reminding us, we have included the data sources of the articles we refer to in the introduction.
Comment 3:
The authors can elaborate on ROS actions on individual breast phenotypes and the underlying mechanisms, including signaling pathways or molecular interactions involved, to enhance the depth of understanding for readers.
Response 3:
Thank you for reminding us, in this article, we summarize the different aspects of ROS affecting the progression of breast cancer, and we also summarize the regulation of related signaling pathways involved.
Comment 4:
Authors could create a comprehensive table presenting the validity of the occurrence, development, and treatment of ROS in breast cancer, encompassing in vitro and in vivo models, drugs used, routes of administration, mechanisms, and outcomes of the studies.
Response 4:
Thank you for reminding us, we have organized these contents into a table based on your proposal.
Comment 5:
Authors may include a list of clinical studies related to oxidative stress employed by drugs or other sources that may be used in the development and treatment of breast cancer.
Response 5:
Thank you for reminding us, we summarized the relevant content in the third part.
Comment 6:
Oxidative stress has detrimental effects on normal cells. Hence, authors may discuss the balance between the benefits of inhibiting cancer cell growth and the need to minimize harm to surrounding healthy tissues.
Response 6:
Thank you for reminding us, we have supplemented this content in the Discussion section.
Comment 7:
The authors can include future research directions and potential study limitations of the present study.
Response 7:
Thank you for reminding us, we have supplemented this content in the Discussion section.
Reviewer 3 Report
none
In this manuscript authors reiewed the current literature regarding the impact of oxidative stress imbalance on the occurrence and progression of breast cancer, elucidating the complex mechanisms by which ROS operate in this context and their therapeutic applications.
The manuscript is interestin, generally well written and illustrated. However, there are some points that deserve to be improved. See my comments below.
Headings and Subheadings must be formatted according to the journal style
Lines 104-105: Since this is a review article and should give an overview of the topic treated, it deserves to be ponted out that oxidative stress is also involved in non-cancerous diseases (see PMID: 39456522 , PMID: 34467607 ).
2. Relationship between oxidative stress and breast cancer: A table summarizing the results of the studies discussed in this section should be added
Abbreviations must be written in full length when mentioned for the first time
An accurate revision of typing errors is suggested
Author Response
In this manuscript authors reviewed the current literature regarding the impact of oxidative stress imbalance on the occurrence and progression of breast cancer, elucidating the complex mechanisms by which ROS operate in this context and their therapeutic applications.
The manuscript is interesting, generally well written and illustrated. However, there are some points that deserve to be improved. See my comments below.
Comment 1:
Headings and Subheadings must be formatted according to the journal style
Response 1:
Thank you for reminding us, we have modified the title style of this article to comply with the journal’s requirements.
Comment 2:
Lines 104-105: Since this is a review article and should give an overview of the topic treated, it deserves to be ponted out that oxidative stress is also involved in non-cancerous diseases (see PMID: 39456522 , PMID: 34467607 ).
Response 2:
Thanks to the reviewer’s comment. To provide a more comprehensive overview, we have added the role of ROS in non-tumor diseases. These two documents are of reference significance. We have cited the documents PMID: 39456522, PMID: 34467607.
Comment 3:
Relationship between oxidative stress and breast cancer: A table summarizing the results of the studies discussed in this section should be added
Response 3:
We thank you for this suggestion. We have summarized the key findings of the studies , and organized it into tables.
Comment 4:
Abbreviations must be written in full length when mentioned for the first time
Response 4:
Thank you for this suggestion, we have ensured that all abbreviations are written in full upon their first use, followed by the abbreviation in parentheses.as required.
Comment 5:
An accurate revision of typing errors is suggested
Response 5:
We appreciate your attention to detail regarding the typos. we have thoroughly checked the full text and corrected all identified typing errors. as required.
Reviewer 4 Report
The manuscript by Dong et al. entitled “Role of oxidative stress in the occurrence, development and treatment of breast cancer” investigates the contribution of oxidative stress to breast cancer. The topic has been already extensively covered in some reviews, including the recently published ones (PMID: 38785550). In general, the manuscript covers multiple aspects of redox biology in breast cancer and is illustrated well. Nevertheless, some issues should be improved before the manuscript can be further considered for publications. Since the title emphasizes that the paper focuses on the oxidative stress, it is suggested to carefully discuss alterations of the antioxidant system observed in breast cancer. In addition, it would be beneficial to summarize the findings of genomic studies revealing driver mutations in redox metabolism-regulating genes.
Major issues:
- It is sufficient to provide an abbreviation with a full name when mentioning for the first time. Thereafter, only the abbreviation should be used. However, all abbreviations should be provided in full.
- Section 1.2. ROS dichotomy and evolutionary importance in regulation of cell energy metabolism and nucleus-mitochondria crosstalk are not covered. More data on their physiological functions should be provided. Moreover, even the pathological aspects of ROS are poorly covered.
- Line 53-55. Provide more specific data on the physiological role of ROS.
- Line 59. Rephrase: NOX is not an organelle
- Line 59-61. Provide specific pathways for ROS generation
- Line 75. GSH is a reduced glutathione. Glutathione peroxidases are normally abbreviated as GPx
- Line 74. SOD is a superoxide dismutase. Specific enzyme names should be provided.
- Section 2. Compared to the previously published review (PMID: 38785550), nothing significant is added.
- Gene names should be italicized
- Links between ROS and ferroptosis are poorly covered
- It is suggested to additionally focus on the role of ROS in regulated cell death modes in breast cancer, including pyroptosis, PANoptosis, or necroptosis. Do ROS contribute to the crosstalk between these RCDs in breast cancer?
Minor issues:
- Remove dots after Subheadings
- Line 59: ROS instead of Ros
- Line 61. The full name for NOX is provided above. No need to do that twice. The same is applied to other abbreviation
- The text contains multiple typos. The authors should read carefully the manuscript to get rid of them
- Make sure that formulas of ROS are properly spelled (subscripts, superscripts, etc.)
Author Response
Comment 1:
The manuscript by Dong et al. entitled “Role of oxidative stress in the occurrence, development and treatment of breast cancer” investigates the contribution of oxidative stress to breast cancer. The topic has been already extensively covered in some reviews, including the recently published ones (PMID: 38785550). In general, the manuscript covers multiple aspects of redox biology in breast cancer and is illustrated well. Nevertheless, some issues should be improved before the manuscript can be further considered for publications. Since the title emphasizes that the paper focuses on the oxidative stress, it is suggested to carefully discuss alterations of the antioxidant system observed in breast cancer. In addition, it would be beneficial to summarize the findings of genomic studies revealing driver mutations in redox metabolism-regulating genes.
Major issues:
Comment 1:
It is sufficient to provide an abbreviation with a full name when mentioning for the first time. Thereafter, only the abbreviation should be used. However, all abbreviations should be provided in full.
Response 1:
Thank you for this suggestion. We have revised the full text as required to ensure that all abbreviations are written out in full on their first use, followed by the abbreviation in parentheses.
Comment 2:
Section 1.2. ROS dichotomy and evolutionary importance in regulation of cell energy metabolism and nucleus-mitochondria crosstalk are not covered. More data on their physiological functions should be provided. Moreover, even the pathological aspects of ROS are poorly covered.
Response 2:
Thank you for your feedback, we have supplemented these contents in Section 1.2.
Comment 3:
Line 53-55. Provide more specific data on the physiological role of ROS.
Response 3:
Thank you for your suggestion. Lines 53-55 have been revised to include more specific data on the physiological role of ROS.
Comments 4:
Line 59. Rephrase: NOX is not an organelle
Response 4:
Thank you for pointing out this inaccuracy. It has been revised to accurately reflect that NOX is not an organelle.
Comments 5:
Line 59-61. Provide specific pathways for ROS generation
Response 5:
Thank you for your suggestion.The process of ROS generation is clarified in line 66-106.
Comments 6:
Line 75. GSH is a reduced glutathione. Glutathione peroxidases are normally abbreviated as GPx
Response 6:
Thank you for pointing out this oversight. This error has been accurately corrected
Comments 7:
Line 74. SOD is a superoxide dismutase. Specific enzyme names should be provided.
Response 7:
Thank you for pointing out this oversight.We have corrected the error.
Comments 8:
Section 2. Compared to the previously published review (PMID: 38785550), nothing significant is added.
Response 8:
Thank you for your comment. While there is some overlap with PMID 38785550, Section 2 has been expanded to include a more detailed discussion of the clinical data and treatment strategies related to oxidative stress in breast cancer.
Comments 9:
Gene names should be italicized
Response 9:
Thank you for this suggestion. All gene names have been italicized throughout the manuscript.
Comments 10:
Links between ROS and ferroptosis are poorly covered
Response 10:
Thank you for your feedback. We have added a new section lines 655-690 detailing the links between ROS and ferroptosis.
Comments 11:
It is suggested to additionally focus on the role of ROS in regulated cell death modes in breast cancer, including pyroptosis, PANoptosis, or necroptosis. Do ROS contribute to the crosstalk between these RCDs in breast cancer?
Response 11:
Thank you for this suggestion. We have reviewed for relevant literature and supplemented the content of combined ROS-mediated cell death regulation for breast cancer treatment.
Minor issues:
Comments 12:
Remove dots after Subheadings
Response 12:
Thank you for pointing this out. All dots after subheadings have been removed.
Comments 13:
Line 59: ROS instead of Ros
Response 13:
Thank you for pointing out this typo. Line 59 has been corrected to read "ROS."
Comments 14:
Line 61. The full name for NOX is provided above. No need to do that twice. The same is applied to other abbreviation
Response 14:
Thank you for your feedback. Redundant full names for abbreviations, such as NOX on line 61, have been removed throughout the manuscript.
Comments 15:
The text contains multiple typos. The authors should read carefully the manuscript to get rid of them
Response 15:
Thank you for pointing out the typos. We have carefully proofread the manuscript and corrected all identified errors.
Comments 16:
Make sure that formulas of ROS are properly spelled (subscripts, superscripts, etc.)
Response 16:
Thank you for this important comment. We have carefully reviewed all chemical formulas for ROS and corrected any errors in subscripts and superscripts.
Round 2
Reviewer 1 Report
The authors have revised the manuscript in accordance with the suggestions provided. I wholeheartedly recommend the publication of this work.
The authors have revised the manuscript in accordance with the suggestions provided. I wholeheartedly recommend the publication of this work.
Author Response
Comments 1: The authors have revised the manuscript in accordance with the suggestions provided. I wholeheartedly recommend the publication of this work.
Response 1: Thanks to you for taking time out of your busy schedules to review this manuscript and provide useful suggestions.
Reviewer 2 Report
Accept in present form
Accept in present form
Author Response
Comments 1: Accept in present form
Response 1: Thanks to your for taking time out of their busy schedules to review this manuscript and provide useful suggestions.
Reviewer 4 Report
The authors have mainly addressed the comments.
However, the following issues should be additionally considered:
- “ROS” is usually plural
-Line 68. Ensure that the superscript is used in the formula of superoxide ion
Line 79. ROS abbreviation is provided in full in Line 52
Line 676. GSH-Px abbreviation is provided in full in Line 113
Line 212. Do ROS regulate or cause the oxidative modification of DNA?
“Minus” in the structural formula of peroxynitrite should be in superscript
Line 260. Use RS instead of replication stress
Line 450 In studies by Li Zhao should be replaced with “In a study by Zhao et al.,”
Author Response
Comments 1: However, the following issues should be additionally considered:
- “ROS” is usually plural
Response 1: Thank you for pointing out this, we have corrected the mistakes in the manuscript.
Comments 2: Line 68. Ensure that the superscript is used in the formula of superoxide ion
Response 2: Thank you for pointing out this oversight. We have corrected the mistake.
Comments 3: Line 79. ROS abbreviation is provided in full in Line 52
Response 3: Thank you for pointing out this oversight. We have corrected the mistake.
Comments 4: Line 676. GSH-Px abbreviation is provided in full in Line 113
Response 4: Thank you for pointing out this oversight. The mistake has been accurately corrected.
Comments 5: Line 212. Do ROS regulate or cause the oxidative modification of DNA?
Response 5: Thank you for pointing out this inaccuracy. We have corrected this to ROS oxidation of dNTPs.
Comments 6: “Minus” in the structural formula of peroxynitrite should be in superscript
Response 6: Thank you for pointing out this oversight. The mistake has been accurately corrected.
Comments 7: Line 260. Use RS instead of replication stress
Response 7: Thank you for pointing out this oversight. We have corrected the mistake.
Comments 8: Line 450 In studies by Li Zhao should be replaced with “In a study by Zhao et al.,”
Response 8: Thank you for pointing out this oversight. We have corrected the mistake.